# From 18- to 20-electron ferrocene derivatives via ligand coordination

Satoshi Takebayashi [1] ✉, Jama Ariai [2,6], Sergey V. Kartashov [3,6], Robert R. Fayzullin [3] ✉, Tomoko Onoue[4], Ko Mibu [4], Hyung-Been Kang [5] & Noriko Ishizu[5]

The 18-electron rule is a fundamental rule in coordination chemistry on which several revolutionary discoveries in catalysis and materials science are founded. This rule has classes of exceptions; however, it is widely taught and accepted that diamagnetic 18-electron complexes do not coordinate to a ligand to form a 20-electron complex even as a reaction intermediate. Here, based on tunable ligand design, we report the formation of 20-electron ferrocene derivatives through reversible nitrogen coordination to 18-electron analogs. Through theoretical studies, we have elucidated key features that enabled this coordination chemistry and how the nitrogen coordination shifts the metal–ligand bonding characters. These 20-electron ferrocene derivatives exhibit reversible $Fe^{II}/Fe^{III}/Fe^{IV}$ redox chemistry under previously unattainable, mild conditions. This work highlights the previously unknown coordination chemistry of diamagnetic 18-electron complexes, which underlies the foundation for future innovations in a range of synthetic chemistry.

The 18-electron rule, proposed in 1921 by Langmuir[1], is a widely taught and used method for predicting the structures and stability of diamagnetic *d*-block coordination complexes[2]. Three Nobel prize discoveries in this century[3–5] and cutting-edge applications, including metal-organic frameworks (MOFs)[6], are founded on this rule to a certain extent. This rule is known to have several classes of exceptions, such as the stability of paramagnetic 20-electron nickelocene[7], and $d^8$ square planar 16-electron complexes. However, the stability of diamagnetic 18-electron complexes, especially complexes with $d^6$ electron configuration, is the commonly accepted experimental foundation of this rule. Against this backdrop, the coordination of a ligand to a diamagnetic 18-electron complex that results in a formal 20-electron complex is generally considered improbable, even as a reaction intermediate[8]. One of the most prominent examples of the 18-electron rule is the structure and stability of ferrocene, which marks the beginning of modern organometallic chemistry[9]. Various derivatives of ferrocene have been synthesized and used in important

applications, such as catalysis[10], solar cell components[11], medicines[12], medical devices[13], and advanced materials[14,15]. The versatility of ferrocene derivatives is attributed mainly to their stability, reversible one-electron oxidation under mild potentials, and ease of functionalization[16,17]. The stability of ferrocene derivatives stems from their coordinatively saturated, formal 18-electron configuration. While reduction or oxidation of ferrocene derivatives to a formal 19-electron anion[18,19] or 16-electron dication[20–22], respectively, requires some of the strongest reductants or oxidants (Fig. 1a), ferrocene derivatives can be oxidized to a formal 17-electron cation at mild oxidation potentials[23,24] since the iron center is relatively electron-rich. For the same reason, ferrocene derivatives can coordinate to Lewis acidic species, such as boranes, and cationic or coordinatively unsaturated metal centers[25–29] (Fig. 1b). In contrast, the coordination of a Lewis basic ligand to iron atom in ferrocene has remained elusive (Fig. 1c). For example, ferrocene does not react with CO even under 150 atm[30]. In theory, this coordination would form formal 20-electron ferrocene derivatives if

[1]Organometallic Chemistry Group, Okinawa Institute of Science and Technology Graduate University, Onna-son, Japan. [2]Institute of Organic Chemistry, Justus Liebig University Giessen, Giessen, Germany. [3]Arbuzov Institute of Organic and Physical Chemistry, FRC Kazan Scientific Center, Russian Academy of Sciences, Kazan, Russian Federation. [4]Graduate School of Engineering, Nagoya Institute of Technology, Nagoya, Japan. [5]Engineering Section, Okinawa Institute of Science and Technology Graduate University, Onna-son, Japan. [6]These authors contributed equally: Jama Ariai, Sergey V. Kartashov. ✉e-mail: satoshi.takebayashi@oist.jp; robert.fayzullin@gmail.com

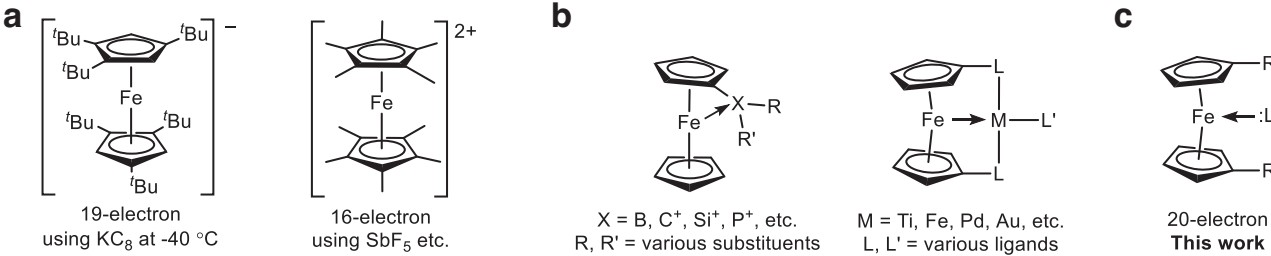

**Fig. 1 | Known and elusive coordination chemistry of 18-electron ferrocene derivatives. a** Reduction and oxidation to 19- and 16-electron species. *t*Bu: *tert*-butyl. **b** Coordination to Lewis acidic species. **c** Previously unknown coordination of a Lewis basic ligand (:L).

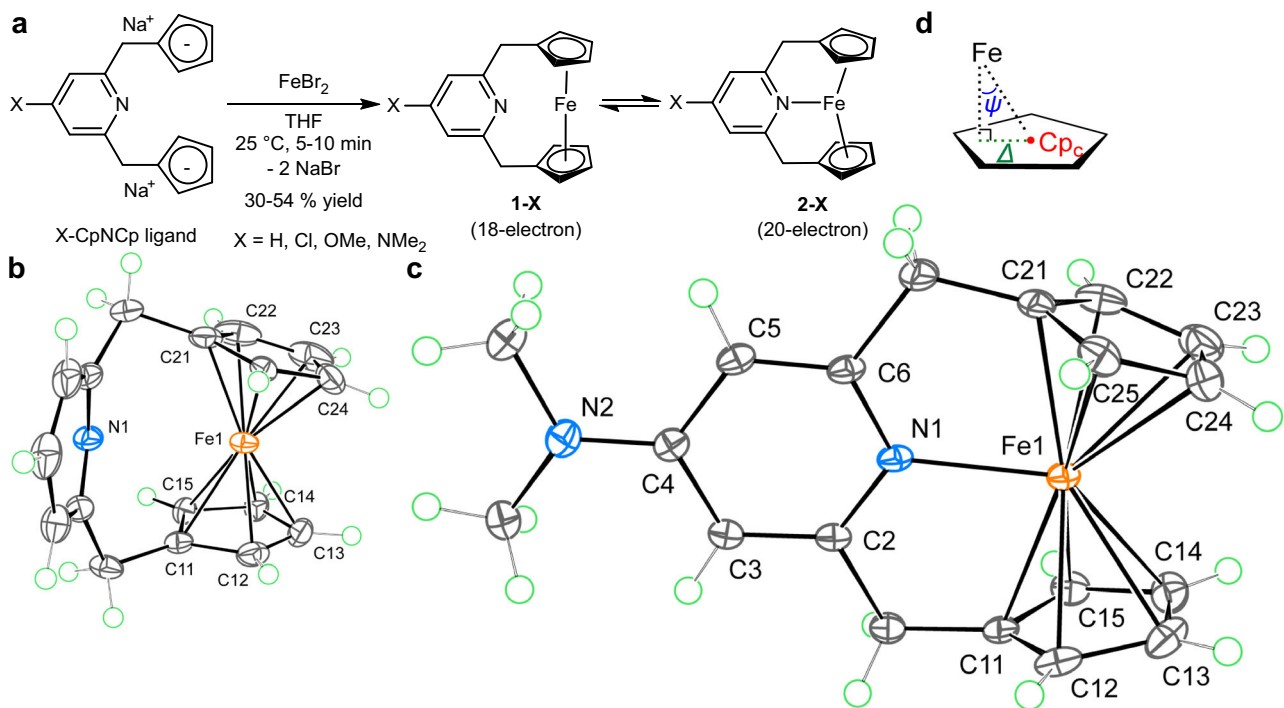

**Fig. 2 | Synthesis and SC-XRD structures of complex 1-X and 2-X. a** Synthetic route to complexes **1-X** and **2-X** (X = H, Cl, OMe, and NMe₂). Me: methyl. **b** X-ray molecular structure of **1-H** with thermal ellipsoids at the 60% probability level; green circles represent hydrogen atoms. **c** X-ray molecular structure of **2-NMe₂** with thermal ellipsoids at the 60% probability level. **d** Geometrical definition of Cp ring slip parameters $\Delta$ and $\psi$ with respect to the Cp ring centroid (Cp_c).

the $\eta^5$-coordination mode of the two cyclopentadienyl (Cp) groups does not change upon the coordination[31]. A bonding model based on $sd^n$ metal hybridization and 3c–4e hypervalent bonding predicts the possibility of such coordination chemistry of 18-electron complexes[32,33]. However, such coordination chemistry of diamagnetic 18-electron ferrocene derivatives is considered unlikely according to the 18-electron rule. Here, we report the synthesis of formal 20-electron ferrocene derivatives by reversible intramolecular coordination of a pyridine ligand to the corresponding diamagnetic 18-electron derivatives. Moreover, this coordination chemistry enables unusually facile stepwise oxidation of the ferrocene derivatives from Fe^II through Fe^III to Fe^IV.

## Results and discussion
### Synthesis and characterization
To observe the improbable coordination chemistry of 18-electron ferrocene derivatives, we developed a simple X-CpNCp ligand system (Fig. 2a) that could stabilize the expected 20-electron species. Our previous study showed that the H-CpNCp ligand promotes nitrogen coordination to a derivative of paramagnetic 19-electron cobaltocene, a typical exception to the 18-electron rule[34]. The addition of the

Na₂[H-CpNCp] salt to a tetrahydrofuran (THF) solution of FeBr₂ resulted in the rapid formation of NaBr precipitate and a dark red-brown solution. After workup, a dark orange crystalline solid of the Fe(H-CpNCp) complex, **1-H**, was isolated in 54% yield. The single-crystal X-ray diffraction (SC-XRD) study revealed that the Fe···N distance in **1-H** is 3.0403(8) Å, and the nitrogen electron pair is directed away from the iron atom, while the two Cp groups are bound to Fe in the $\eta^5$-coordination mode (Fig. 2b). Therefore, in the crystalline phase, **1-H** is a formal 18-electron complex without an Fe–N bond, reflecting the expected stability of an 18-electron complex. Consistent with this observation, the ⁵⁷Fe Mössbauer spectrum of the solid sample of **1-H** at 77 K (Fig. 3a, c) showed a doublet signal with isomer shift (δ) and quadrupole splitting ($\Delta E_Q$) similar to ferrocene[35] (Fig. 3c), and solid-state magnetic measurements by vibrating sample magnetometer (VSM) supported the formation of a diamagnetic complex. However, in solution, extremely broad ¹H nuclear magnetic resonance (NMR) signals were observed at 25 °C (Supplementary Fig. 7). Variable-temperature NMR (from −40 °C to 150 °C) and solution magnetic measurements by Evans' method showed the presence of a temperature-dependent equilibrium between diamagnetic and paramagnetic species. Consistent with the solid-state structure, the

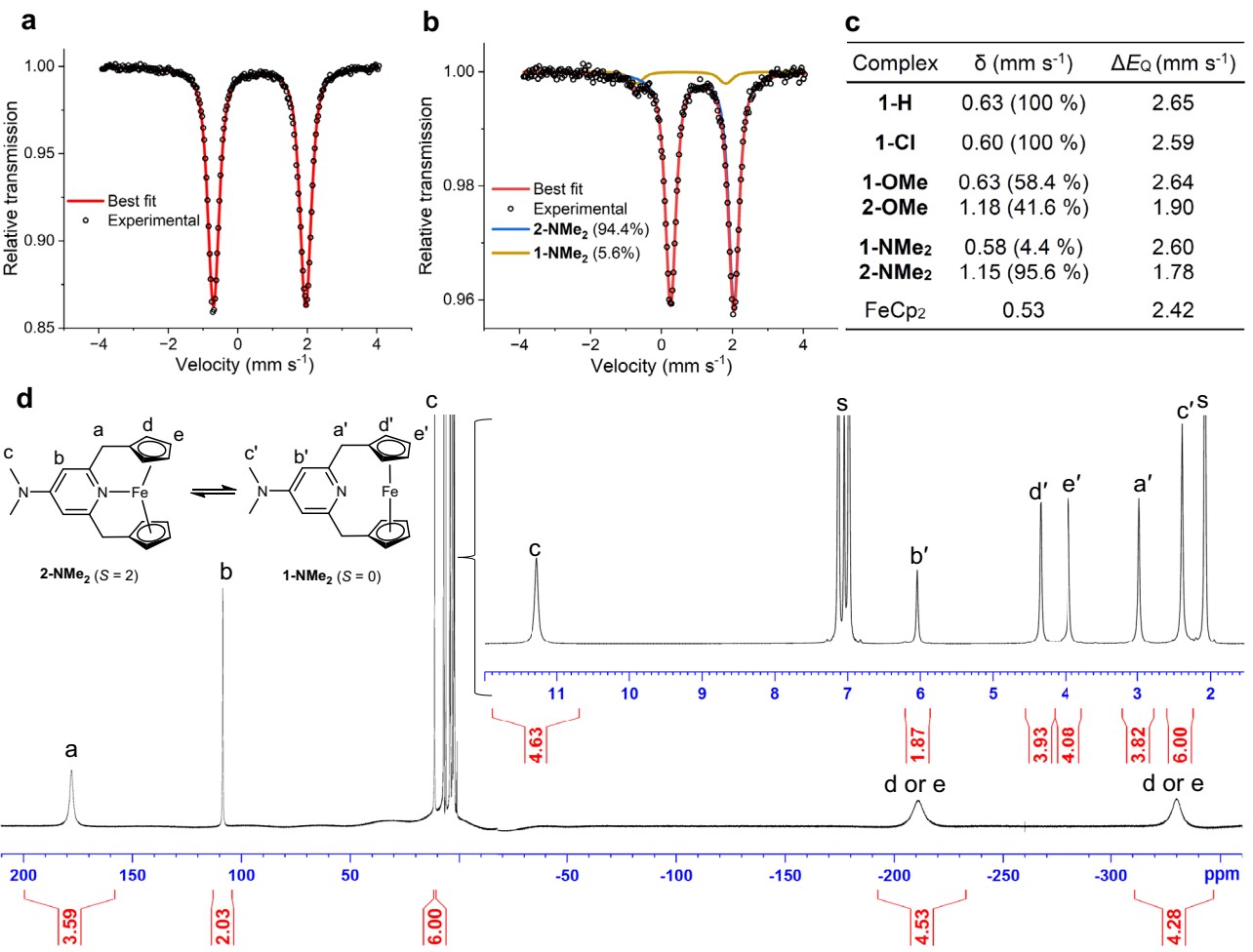

**Fig. 3 | Characterization of 1-X and 2-X. a** $^{57}$Fe Mössbauer spectrum of **1-H** recorded at 77 K. **b** $^{57}$Fe Mössbauer spectrum of **2-NMe$_2$** recorded at 77 K. **c** Fitting parameters for the $^{57}$Fe Mössbauer spectra of **1-X** and **2-X** at 77 K and ferrocene (FeCp$_2$) at 90 K[35]. **d** $^1$H NMR spectra (500.13 MHz, −40 °C) of an equilibrium mixture of **1-NMe$_2$** and **2-NMe$_2$** in toluene-$d_8$, as well as corresponding signal assignments (a–e and a′–e′). Two separate spectra recorded between 380 ppm to −20 ppm and between 20 ppm to −380 ppm were combined to show the entire region with accurate baseline corrections and integration values (see Supplementary Fig. 25 for an uninterrupted spectrum). s: Residual solvent signals from toluene-$d_8$.

diamagnetic species observed at −40 °C showed a $^{15}$N NMR chemical shift indicating the absence of an Fe−N bonding interaction (Supplementary Fig. S9). Whereas the $^1$H NMR spectrum of the paramagnetic species recorded at 150 °C supported the $C_{2v}$ molecular symmetry and indicated formation of *N*-coordination species.

Aiming at isolating Fe-N bonded species, we synthesized the X-CpNCp ligands with varying Lewis basicity (Fig. 2a). The Lewis acid–base interaction between the metal and the pyridine group was previously identified as the key driving force for the coordination of a pyridine group to the cobaltocene derivative[34]. When most Lewis basic NMe$_2$-CpNCp ligand was reacted with FeBr$_2$, the *N*-coordinated complex **2-NMe$_2$** was isolated as an off-white solid in 39% yield. Complex **2-NMe$_2$** is air sensitive; however, it is stable under a nitrogen atmosphere at 25 °C for more than a year. VSM measurements confirmed the formation of a high-spin, $S = 2$ complex with an effective magnetic moment of 4.9 μ$_B$ (μ$_B$: Bohr magneton) at 298 K. The SC-XRD structure of **2-NMe$_2$** (Fig. 2c) revealed a significantly shortened Fe−N distance of 2.1476(10) Å, which is among the shortest Fe$^{II}$−N distances in high-spin, pyridine-based Fe$^{II}$PNP pincer complexes (Supplementary Table 12). Fe···Cp centroid (Fe···Cp$_c$) distances of 2.0383(6) Å and 2.0393(5) Å are more than 0.33 Å longer than those of **1-H** (1.689(4)−1.709(2) Å) and ferrocene (1.651 Å)[36], and represent the longest Fe···Cp$_c$ distances in Fe$^{II}$ ferrocene derivatives[37], suggesting a substantial weakening of Fe···C(Cp) interatomic interactions. The coordination mode of the Cp

groups was analyzed by the Cp ring slip parameters $\Delta$ and $\psi$ (Fig. 2d)[38]. $\Delta$ distances of 0.112 Å and 0.124 Å and $\Psi$ angles of 3.2° and 3.5° for the two Cp groups of **2-NMe$_2$**, respectively, fall within the range that supports the $\eta^5$-coordination mode[38,39]. In agreement with this analysis, the $\Delta$ and $\Psi$ values of $\eta^5$-coordinated Cp groups in **1-H** were 0.074−0.252 Å and 2.5° to 5.9°, respectively. Unlike other ferroceno-phane derivatives[40], the Cp rings in **2-NMe$_2$** are tilted toward the bridging moiety of the ligand, forming a Cp$_c$···Fe···Cp$_c$ angle of 145.92(3)°. This significant inward tilting corroborates the presence of an attractive interaction between the Fe and N atoms. In contrast, the bending of the Cp$_c$···Fe···Cp$_c$ angle was much smaller in **1-H** (about 170.6°). The $^{57}$Fe Mössbauer spectrum of solid **2-NMe$_2$** at 77 K (Fig. 3b, c) shows a doublet signal at $\delta = 1.15$ mm s$^{-1}$ with a trace doublet signal from *N*-noncoordinated complex **1-NMe$_2$** at $\delta = 0.58$ mm s$^{-1}$. Note that $\delta$ values more than 1 mm s$^{-1}$ are characteristic of $S = 2$, Fe$^{II}$ complexes[41] and consistent with the spin and oxidation state of **2-NMe$_2$**.

In solution, the $^1$H NMR spectrum recorded at −40 °C showed five broad signals between −330 ppm and +180 ppm, consistent with the formation of paramagnetic **2-NMe$_2$**. In addition to these signals, five sharper signals appeared in the typical diamagnetic region (0–10 ppm) of the spectrum (Fig. 3d). Based on two-dimensional (2D) NMR experiments, selective deuteration of Cp protons, and $^1$H NMR integral ratios, these signals in the paramagnetic and diamagnetic regions were

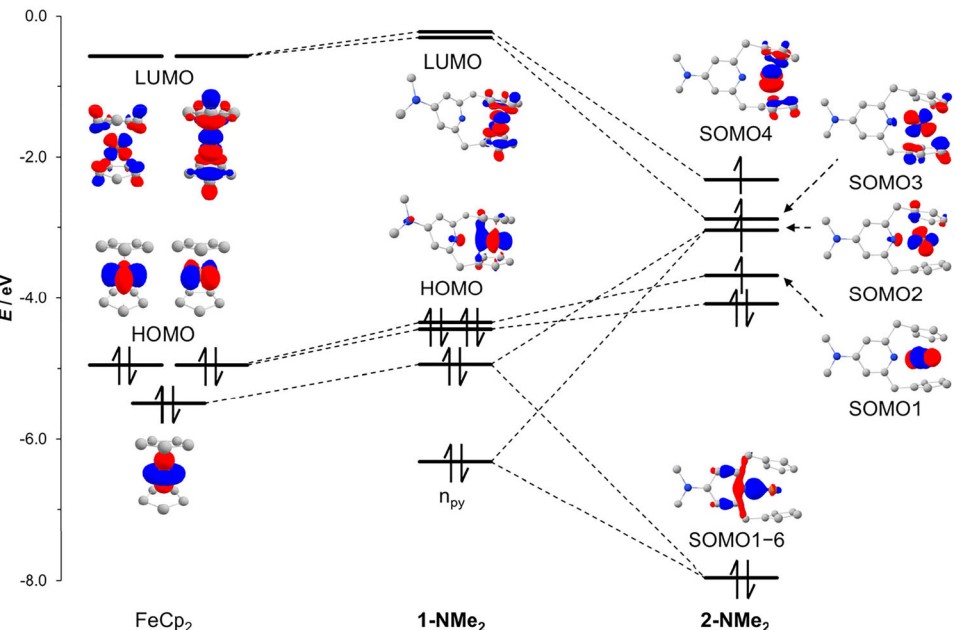

**Fig. 4 | Comparison of MO diagrams of ferrocene (FeCp₂), 1-NMe₂, and 2-NMe₂.** The calculations were carried out at the TPSSh-D4/def2-QZVPP//TPSS-D4/def2-TZVPP level of theory. Inset: canonical KS-MOs (isovalues are at 0.05–0.07 a.u.), blue lobes: positive phases, and red lobes: negative phases. For a detailed MO diagram and KS-MOs, please see Supplementary Figs. 65–67. HOMO highest occupied MO, LUMO lowest unoccupied MO, SOMO singly occupied MO, SOMO1–6 an MO at six energy levels below SOMO1, $n_{py}$: nitrogen lone pair.

assigned to *N*-coordinated **2-NMe₂** and *N*-noncoordinated **1-NMe₂**, respectively. Notably, **1-NMe₂** displays a $^{15}$N NMR signal of the pyridine nitrogen at 299 ppm (233 K, in toluene-$d_8$), similar to the chemical shift of the sodium salt of the free ligand at 262 ppm (233 K, in toluene-$d_8$ with THF-$d_8$). The ratio of **1-NMe₂** to **2-NMe₂** varies with temperature, and the exchange between the two species was confirmed by 2D $^1$H-$^1$H exchange (EXSY) NMR measurements (Supplementary Fig. 32). Therefore, we experimentally observed reversible formation of the formal 20-electron **2-NMe₂** via the coordination of nitrogen to the 18-electron **1-NMe₂**. Thermodynamic parameters for the formation of **1-NMe₂** from **2-NMe₂** ($\Delta H^o = -1.98(3)$ kcal mol$^{-1}$, $\Delta S^o = -8.1(1)$ cal mol$^{-1}$, and $\Delta G_{298} = 0.44(4)$ kcal mol$^{-1}$ in toluene-$d_8$) were determined using a van't Hoff plot, indicating that high-spin **2-NMe₂** is entropically favored, similar to other spin-crossover iron complexes[42]. The equilibrium concentration of **2-NMe₂** is higher in more polar solvents (CD₃CN > THF-$d_8$ > toluene-$d_8$) at 230.2 K (Supplementary Fig. 33), likely due to the greater polarity and ionic character of the Fe···C(Cp) bonds (see below), which result in a higher molecular dipole moment of **2-NMe₂** (8.3 D compared to 4.1 D for the free **2-NMe₂** and **1-NMe₂**, based on theoretical computations). Interestingly, when the less Lewis basic OMe-CpNCp ligand was used, *N*-noncoordinated **1-OMe** and *N*-coordinated **2-OMe** were observed as a cocrystal in a 1:1 ratio in the asymmetric cell by SC-XRD (Supplementary Fig. 72). Consistent with this observation, the $^{57}$Fe Mössbauer spectrum showed two sets of doublet signals corresponding to **1-OMe** and **2-OMe** in a 58.4:41.6 ratio, respectively (Supplementary Fig. 56a). When the least Lewis basic Cl-CpNCp ligand was used, only *N*-noncoordinated **1-Cl** was obtained in the solid state. However, in solution, variable temperature NMR indicated the presence of reversible *N*-coordination. Therefore, in solution, the coordination of the pyridine groups to the 18-electron ferrocene derivative is reversible, and the strength of coordination is tunable by the Lewis basic character of the pyridine groups. Theoretical calculations also predicted this tunability of the nitrogen coordination and showed that it depends on the interplay of the Lewis basicity of the pyridine ligand, entropic effects caused by the **1-X/2-X** transformation, and stabilizing solvent effects.

## Computational study

The electronic structure of **2-NMe₂** was further investigated by density functional theory (DFT) calculations at the TPSSh-D4/def2-QZVPP//TPSS-D4/def2-TZVPP level of theory[43–48]. Consistent with the experimental observation, the calculated structure of **2-NMe₂** showed long Fe···Cp$_c$ distances (2.017 and 2.039 Å) and a short Fe−N distance (2.1853 Å) (Supplementary Table 2). The optimized geometry of **1-NMe₂** is characterized by an Fe···N internuclear distance of 3.0540 Å and Fe···Cp$_c$ distances of 1.656 Å and 1.662 Å. In addition, the calculations with the implicit solvent model[49] predict that **2-NMe₂** should be energetically favored over **1-NMe₂** as the polarity of the solvent increases (Supplementary Table 4), which is in accordance with the NMR experiments. Furthermore, in agreement with the experimental results, the calculated thermodynamic contributions reveal that **2-NMe₂** is indeed entropically favored over **1-NMe₂**. This is mainly due to the contributions of vibrational entropy originating from longer Fe···C(Cp) distances (Supplementary Table 5). The frontier molecular orbital (MO) diagrams of ferrocene (FeCp₂), **1-NMe₂**, and **2-NMe₂**, derived from canonical Kohn–Sham quasi-restricted molecular orbitals (KS-MOs)[50], are presented in Fig. 4. The MO diagrams indicate that ferrocene and **1-NMe₂** share a similar orbital arrangement, whereas significant reorganization of the MOs is observed in **2-NMe₂** as expected from the bending of the Cp···Fe···Cp fragment[51]. The canonical KS-MOs of **2-NMe₂** reveal a σ-type interaction between the formerly nonbonding $d_{z^2}$-orbital and the nitrogen electron pair (Fig. 4, SOMO1–6; SOMO stands for singly occupied MO). The formation of the Fe−N bond results from the fully occupied Fe−N bonding (SOMO1–6) and half-occupied Fe−N antibonding (SOMO2) orbitals, giving the Fe−N bond in **2-NMe₂** a formal bond order of 0.5. The formation of this bond is associated with the bending of the Cp···Fe···Cp fragment, which significantly stabilizes the orbitals descended from the lowest unoccupied MO (LUMO) of ferrocene (Fig. 4, SOMO3 and SOMO4)[51]. The half-occupation of these Fe···Cp antibonding orbitals explains the significant elongation of Fe···Cp$_c$ distances. This situation is similar to the 20-electron nickelocene[52]. Therefore, the stability of **2-NMe₂** originates from the formation of the Fe−N bond and bending

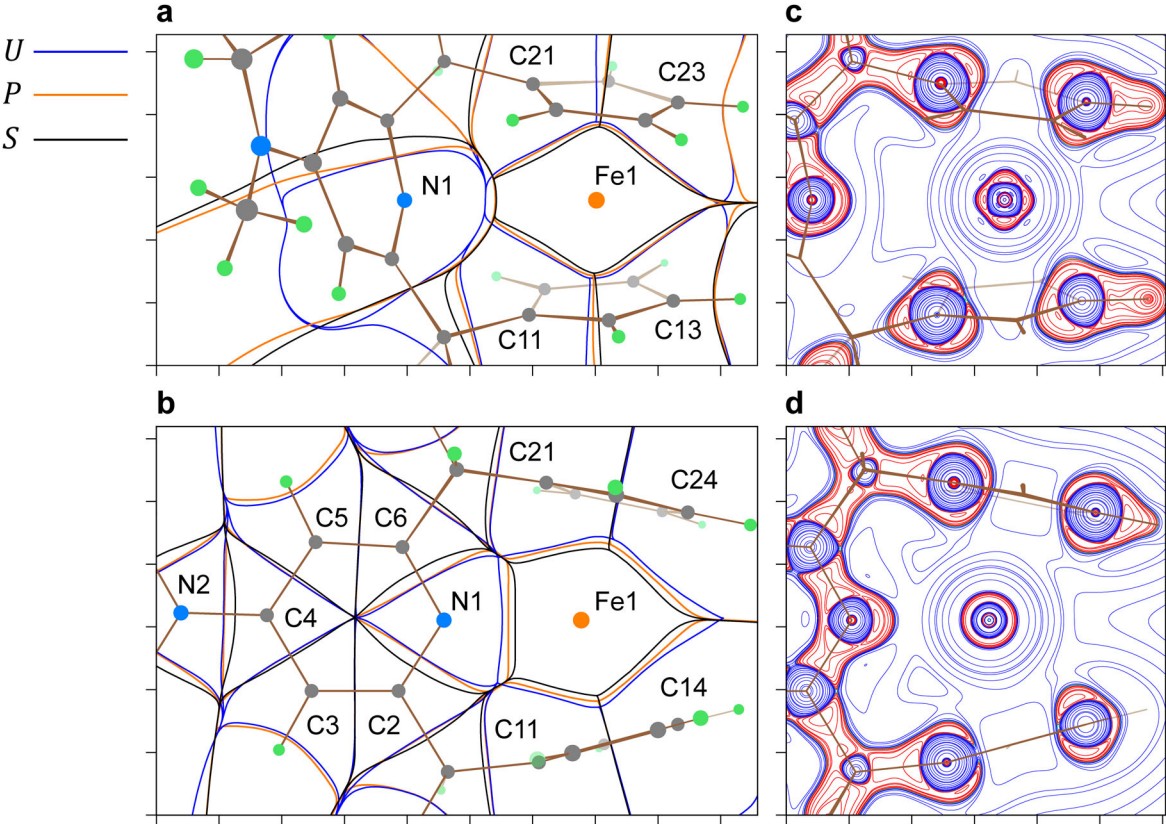

**Fig. 5 | Quantum chemical topology of 1-NMe₂ and 2-NMe₂.** Superpositions of the zero-flux surfaces determined in $\mathbf{F}_{es}(\mathbf{r})$ ($U$, blue), $\mathbf{F}_k(\mathbf{r})$ ($P$, orange), and $\nabla\rho(\mathbf{r})$ ($S$, black) for **a 1-NMe₂** and **b 2-NMe₂**, based on theoretical data. Green dots represent hydrogen atoms. Contour maps of $\nabla^2\rho(\mathbf{r})$ for **c 1-NMe₂** and **d 2-NMe₂** are depicted using a logarithmic scale of $\pm 1 \times 10^n$, $\pm 2 \times 10^n$, $\pm 4 \times 10^n$, and $\pm 8 \times 10^n$ a.u. ($-2 \le n \le 3$). Blue and red colors correspond to positive and negative function values, respectively. The map planes intersect the Fe1, N1, and C21 nuclei. The distance between adjacent axis tick marks is 1 Å.

of the Cp⋯Fe⋯Cp fragment, facilitated by the structure of the NMe₂-CpNCp ligand. Indeed, an intermolecular reaction between ferrocene and 4-dimethylaminopyridine showed no sign of the Fe−N bond formation, even at 130 °C.

## Quantum chemical topology and interatomic interactions

The interatomic interactions in **1-NMe₂** and **2-NMe₂**, as well as the origin of the stability of the 20-electron ferrocene derivatives, were examined by analyzing the relationships between the zero-flux surfaces $U$, $P$, and $S$ defined in the electrostatic force $\mathbf{F}_{es}(\mathbf{r})$, the kinetic force $\mathbf{F}_k(\mathbf{r})$, and the electron density gradient $\nabla\rho(\mathbf{r})$, respectively (Fig. 5a, b)[53–55]. The coordination of N1 by Fe1 leads to a significant reorganization of these zero-flux surfaces. Upon the coordination, a notable increase in metal-to-ligand charge transfer is observed, rising from 0.796 e to 1.123 e. This increase is reflected in the expansion of the gaps between $S$(Fe1|L) and $U$(Fe1|L), where L refers to an atom in the first coordination sphere. In addition to the expansion of the charge transfer region, this reorganization involves $P$(Fe1|C_Cp) moving away from $U$(Fe1|C_Cp), while $P$(Fe1|N1) moving closer to $U$(Fe1|N1). As previously demonstrated, the approach of $P$ toward $U$ within an internuclear region reflects an increase in the electron sharing of the transferred density for a polar interatomic interaction, thereby imparting a more covalent character, and vice versa[56,57]. Thus, the observed arrangement of $U$, $P$, and $S$ indicates that, upon the transition from **1-NMe₂** to **2-NMe₂**, the Fe1−N1 chemical bond forms with substantial facilitation by the electron exchange correlation effect. Concurrently, the transition to the 20-electron configuration is accompanied by a shift in the mode of Fe⋯Cp binding from a predominantly covalent to a more electrostatically governed "ionic" mechanism. This conclusion is further supported by the enhancement

of the internuclear static potential barriers, which inhibit electron delocalization between pairs of atoms[58,59], from −1.075(43) to −0.816(78) a.u., and the decrease in the delocalization indices[60,61] from 0.455(43) to 0.240(64) a.u. for the Fe⋯Cp binding upon the transition (Supplementary Tables 9 and 10). Concurrently, following the formation of the Fe1−N1 bond, the barrier height between Fe1 and N1 diminishes markedly from −0.407 a.u. to −0.917 a.u., while the delocalization index dramatically increases from 0.053 a.u. to 0.364 a.u. On the basis of the same characteristics (Supplementary Table 11), the formation of the analogous Co^II−N coordination bond[34] likewise induces a discernible attenuation of the covalent character of the Co⋯C(Cp) interatomic interactions. Furthermore, the analysis of the Laplacian of the electron density, $\nabla^2\rho(\mathbf{r})$, reveals a pronounced structural distinction between **1-NMe₂** and **2-NMe₂** (Fig. 5c, d). In both cases, charge depletion is observed in the outer shell of the Fe1 atom, associated with the 4$s$ level. In **2-NMe₂**, the formation of the Fe−N bond, concomitant with the enhanced noncovalent character of the Fe⋯Cp binding, is accompanied by the loss of the structured pattern of valence-shell charge concentration at Fe1, attributed to its 3$d$ electrons. This phenomenon indicates reduced directionality of chemical bonding within the coordination sphere of **2-NMe₂**, relative to **1-NMe₂**.

## Electrochemical study

Intrigued by the presence of the high-lying, half-filled Fe⋯Cp antibonding orbitals, we investigated the redox property of the Fe(X-CpNCp) complexes using electrochemical methods. Unexpectedly, the cyclic voltammograms (CVs) of all the Fe(X-CpNCp) complexes (Fig. 6a, b) showed two reversible oxidation events at approximately −0.8 V and 0.1 V vs FeCp₂^{0/+} in THF or CH₂Cl₂ at 23 °C. Ferrocene derivatives are known to undergo reversible one-electron oxidation to

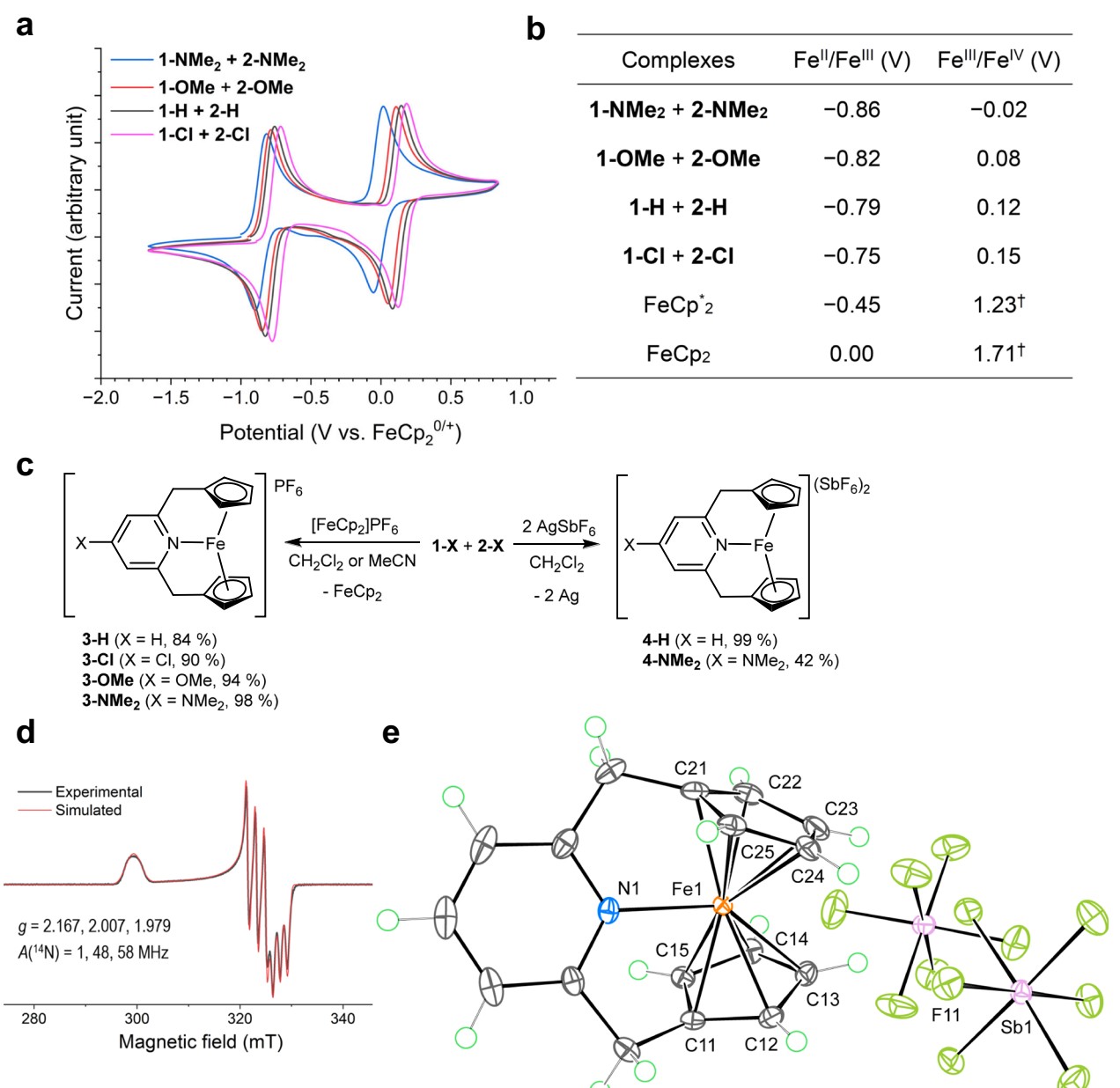

**Fig. 6 | Electrochemical studies. a** CVs (vs FeCp₂$^{0/+}$) of 2.5 mM equilibrium mixtures of **1-X** and **2-X** in 0.2 M NBu₄PF₆ in THF at 23 °C, recorded at a scan rate of 0.1 V s⁻¹. **b** A list of the half-wave potentials; †values obtained in SO₂ at −40 °C[20]. FeCp₂: ferrocene, FeCp*₂: decamethylferrocene. **c** Schematic representation of the chemical oxidation of **1-X** and **2-X**. **d** X-band EPR spectrum of **3-Cl**, recorded in a 1:1 toluene/acetone glass at 77 K. *g*: effective *g*-values, *A*(¹⁴N): ¹⁴N hyperfine splitting constants. **e** X-ray structure of **4-H** with thermal ellipsoids at the 60% probability level; green circles represent hydrogen atoms.

Fe$^{III}$ ferrocenium cations[23], but further oxidation to dicationic Fe$^{IV}$ species requires more electron-rich Cp derivatives such as pentamethylcyclopentadienyl (Cp*) and use of some of the strongest oxidizing reagents[21,22,62,63], due to the prohibitively high oxidation potential and instability of the Fe$^{IV}$ ferrocene dication[20]. The species formed by the two reversible oxidations of the Fe(X-CpNCp) complexes were isolated by chemical oxidations. One-electron oxidation of the Fe(X-CpNCp) complexes with [FeCp₂]PF₆ produced the *S* = 1/2, Fe$^{III}$ complexes: **3-X** (Fig. 6c). Examination of their SC-XRD structures (Supplementary Figs. 75–78) revealed that all the Fe$^{III}$ complexes exhibit the nitrogen electron pairs directed toward the iron nuclei while maintaining the $\eta^5$-coordination mode of the Cp groups. Consistent with the DFT calculations (Supplementary Table 2), the Fe−N distances for **3-X** (2.3785(16)−2.449(3) Å) are significantly longer than that of **2-NMe₂** (2.1476(10) Å), while the Fe···Cp$_c$ distances

(1.7505(9)−1.7582(9) Å) are significantly shorter than that of **2-NMe₂** (2.0383(6) Å and 2.0393(5) Å). The elongation of the Fe−N bonds is explained by the presence of an unpaired electron at an orbital with Fe−N antibonding character. Whereas the shortening of Fe···Cp$_c$ distances are due to the absence of electrons in the Fe···C antibonding orbitals, unlike in the case of **2-NMe₂** (Fig. 4). This explanation was further supported by the presence and absence of spin density at the pyridyl nitrogen atoms and Cp carbon atoms, respectively (Supplementary Fig. 64). Additionally, the electron paramagnetic resonance (EPR) spectra of all the Fe$^{III}$ complexes showed similar patterns, with pronounced ¹⁴N hyperfine splitting features (Fig. 6d and Supplementary Fig. 42). This observation is consistent with the presence of spin density on the pyridyl nitrogen atoms, and bonding interactions between the Fe and N atoms. Therefore, **3-X** are formal 19-electron, complexes.

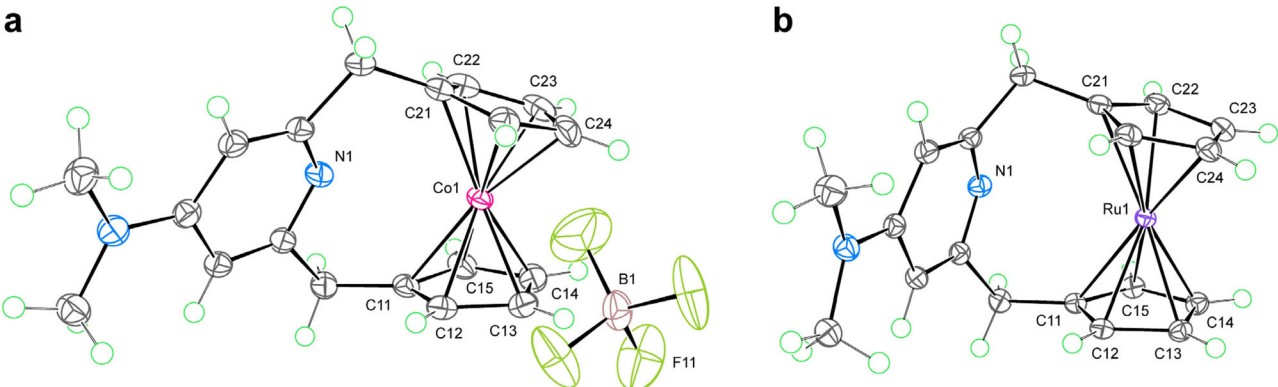

**Fig. 7 | X-ray structures of Co and Ru analogs.** X-ray structures of **a** [Co-NMe$_2$]BF$_4$ and **b** Ru-NMe$_2$ with thermal ellipsoids at the 60% probability level; green circles represent hydrogen atoms.

To our surprise, two-electron oxidation of **1·H** and **2·NMe$_2$** with two equivalents of AgSbF$_6$ formed $S = 0$, Fe$^{IV}$ complexes: **4·H** and **4·NMe$_2$** (Fig. 6c). Therefore, Fe$^{III}$ complexes **3·X** can be oxidized to Fe$^{IV}$ at oxidation potentials near 0 V (Fig. 6b). In the crystals, **4·H** and **4·NMe$_2$** exhibit short Fe–N distances of 2.0437(14) Å and 2.029(5) Å, respectively (Fig. 6e and Supplementary Fig. 80). Fe···Cp$_c$ distances of **4·H** and **4·NMe$_2$** (1.732(3)–1.740(3) Å) are also slightly shorter than those of **3·H** and **3·NMe$_2$**, respectively. The short Fe–N and Fe···Cp$_c$ distances induce significant twisting of the X-CpNCp ligands and increased bending of the Cp···Fe···Cp fragment. However, the $\eta^5$-coordination mode of the Cp groups remains unchanged, supporting the 18-electron configuration of the Fe$^{IV}$ complexes. Notably, oxidation of **2·NMe$_2$** with 1 equiv. [FeCp$_2$]PF$_6$ resulted in a mixture of the corresponding Fe$^{III}$ and Fe$^{IV}$ complexes, **3·NMe$_2$** and **4·NMe$_2$-PF$_6$**, due to the combination of the low solubility of **4·NMe$_2$-PF$_6$** and the Fe$^{III/IV}$ oxidation potential of **2·NMe$_2$** (−0.02 V vs FeCp$_2$$^{0/+}$), which is lower than the Fe$^{II/III}$ oxidation potential of FeCp$_2$. The physical oxidation states of iron atoms in **3·H** and **4·H** were examined using $^{57}$Fe Mössbauer spectroscopy, X-ray photoelectron spectroscopy (XPS), and SC-XRD. The $^{57}$Fe Mössbauer spectra of **3·H** and **4·H** (Supplementary Fig. 57) showed a progressive decrease in isomer shifts, 0.68 mm s$^{-1}$ and 0.48 mm s$^{-1}$, respectively, consistent with the physical oxidation of iron atoms[64,65]. Consistent with this observation, the XPS spectra of **1·H**, **3·H**, and **4·H** (Supplementary Fig. 58) showed an increase in Fe $2p_{3/2}$ binding energy from 707.6 through 708.4 eV to 709.0 eV upon oxidation. Finally, examination of C–C bond distances in the pyridine moieties of **1·H**, **3·H**, and **4·H** showed no sign of oxidation of the pyridine groups (Supplementary Table 13). Altogether, these observations strongly support the physical Fe$^{IV}$ oxidation states of **4·H** and **4·NMe$_2$**. Thus, the formation of the Fe–N bond expands the accessible oxidation states of ferrocene derivatives, which are among the most versatile organometallic redox reagents[66]. We expect this new redox chemistry of ferrocene derivatives will expand its use as a redox mediator or catalyst.

### Coordination chemistry of other 18-electron analogs

The coordination chemistry of the NMe$_2$-CpNCp ligand was further examined using other 18-electron complexes. First, we examined the coordination chemistry of a cationic 18-electron cobaltocenium derivative **[Co-NMe$_2$]BF$_4$** (Fig. 7a). Consistent with the previous study with H-CpNCp ligand[34], solid and solution state characterizations of **[Co-NMe$_2$]BF$_4$** showed the absence of Co–N bonding interaction. Synthesis of Ru analog of **2·NMe$_2$** was attempted using [Ru(*p*-cymene)Cl$_2$]$_2$, however, this reaction also resulted in the formation of a Ru$^{II}$ complex without Ru–N bonding interaction, **Ru-NMe$_2$** (Fig. 7b). These findings suggest that coordination to an 18-electron metallocene is less probable when there are strong metal···Cp bonding interactions, such as

those observed in the cationic and the 2nd row transition metal complexes. The strong bonding interactions result in metal–ligand antibonding orbitals being less accessible to accept additional electrons. In summary, if an 18-electron complex can form low-lying metal–ligand antibonding orbitals upon coordination to a ligand, it may be able to exhibit coordination chemistry similar to that observed in this study. It can be hypothesized that this type of coordination chemistry is most likely to occur in neutral 1st row transition metal complexes. The hypothesis is based on the premise that these complexes tend to have weaker metal–ligand bonding interactions and thus have low-lying metal–ligand antibonding orbitals. Stronger bonding interaction between the incoming ligand and the metal will further facilitate the formation of formal 20-electron species by creating a half-filled, high-lying metal–incoming ligand antibonding orbital. Furthermore, this coordination chemistry will be entropically favored, as it pertains to the weakening of the existing metal–ligand bonding, thereby increasing the metal–ligand vibrational entropy. In typical reaction conditions, the weakening of metal–ligand bonding is likely to result in dissociation and substitution of the ligand. However, under circumstances that preclude dissociation of a ligand, such as those engendered by the ligand design or by the presence of excess dissociating ligand, as observed in certain catalytic reactions, this type of coordination chemistry may occur. A potential reaction example that aligns with the specified reaction conditions is the Co$_2$(CO)$_8$-catalyzed hydroformylation reaction under high temperature and high CO pressure. It has been previously established that, under elevated CO pressure, this catalytic reaction occurs through a mechanism that does not involve the dissociation of CO from Co$_2$(CO)$_8$[67].

Contrary to the conventional knowledge of the 18-electron rule, we have shown that diamagnetic 18-electron ferrocene derivatives can coordinate an additional ligand, forming formal 20-electron ferrocene derivatives. This transformation is driven by the weakening of the Fe···Cp interaction, which results in a lower-lying Fe···Cp antibonding orbital, and is accompanied by a shift in the mode of Fe···Cp binding from predominantly covalent to more ionic. The partial occupation of these high-lying antibonding orbitals enables the facile two-electron oxidation of ferrocene derivatives. This study expands our understanding of 18-electron ferrocene chemistry and the 18-electron rule and their potential applications in synthetic chemistry

## Methods

### General considerations and materials for synthetic study

All metal complexes and ligands were prepared under an N$_2$ atmosphere using an MBRAUN glovebox, UNILAB Plus SP, equipped with an MB-LMF-2/40-REG regenerable solvent trap, an MB-20-G gas purifier, and an MB-GS-35−35 °C freezer. Precursors of ligands were synthesized outside the glovebox using an argon Schlenk line or

under air. All glassware was dried at 170 °C overnight and cooled down to room temperature under a vacuum in the glovebox mini antechamber. All solvents were reagent grade or higher. Hexamethyldisiloxane (Sigma-Aldrich 205389-500 ML, ≥98%), *n*-heptane (≥99.0%), hexanes (≥95.0%, n-hexane with minor amount of isomers of methylpentane, and methylcyclopentane), n-pentane (≥99.0%), toluene (≥99.8%), benzene (≥99.7%), dichloromethane (≥99.5%), THF (Sigma-Aldrich 401757-1L, anhydrous, ≥99.9%, inhibitor-free), diethyl ether (≥99.8%, inhibitor-free), acetonitrile (≥99.8%), methanol (≥99.8%), dimethyl formamide (DMF, ≥99.0%), 1,2-difluorobenzene (TCI, ≥98% or Oakwood Chemical, 99%), and nitromethane (TCI, ≥98%) were kept for more than 2 days with MS3A in the glovebox and kept in the glovebox. MS3A was dried overnight in a 200 °C oven and cooled to room temperature under vacuum overnight in the glovebox large antechamber. Common chemicals were purchased and used as received unless stated otherwise. Anhydrous $FeBr_2$ (brown powder, 98%, 400831-10G) was purchased from Sigma Aldrich. 2,6-Bis(methylenecyclopentadienyl)pyridine disodium salt (**Na₂CpNCp**) was prepared according to a published method[34,68]. Celite filtration was carried out using a pipette, cotton wool, and Celite®545. Celite®545 was dried overnight in a 170 °C oven, cooled to room temperature under vacuum overnight in the glovebox large antechamber, and kept in the glovebox. The dimensions of the 20 mL vial are 28 mm outer diameter and 60 mm in height, and the Teflon-coated stirring bar is 5 mm in diameter and 15 mm long.

### Instrumental analysis methods

**NMR spectroscopy.** NMR spectra were recorded using a Bruker Avance NEO 500 spectrometer equipped with a cryoprobe and an Avance III HD 400 N spectrometer. Deuterated solvents were degassed using argon, dried over dry MS3A for at least 2 days in the glovebox, and stored in the glovebox. $^1H$ and $^{13}C$ NMR chemical shifts are reported in parts per million (δ) relative to TMS (0 ppm) signal with the residual solvent signal ($CDCl_3$: 7.26 ($^1H$) and 77.16 ($^{13}C$) ppm, $C_6D_6$: 7.16 ($^1H$) and 128.06 ($^{13}C$) ppm, toluene-$d_8$: 2.08 ($^1H$) and 20.43 ($^{13}C$) ppm, THF-$d_8$: 1.72 ($^1H$) and 25.31 ($^{13}C$) ppm, DMSO-$d_6$: 2.50 ($^1H$) and 39.52 ($^{13}C$) ppm, methanol-$d_4$: 3.31 ($^1H$) and 49.00 ($^{13}C$) ppm, $CD_3CN$: 1.94 ($^1H$) and 118.26 ($^{13}C$) ppm, $CD_3NO_2$: 4.33 ($^1H$) and 63.0 ($^{13}C$) ppm) as the internal references. $^{11}B$, $^{15}N$, and $^{19}F$ NMR chemical shifts are reported in parts per million (δ) relative to $BF_3 \cdot OEt_2$ in $CDCl_3$ (0 ppm), $NH_3$(liquid) (0 ppm), and $CFCl_3$ (0 ppm), respectively, as the external references. $^2H$ NMR chemical shifts are presented in parts per million (δ) with respect to the signal from the residual solvent as the internal reference.

NMR signal assignments of diamagnetic organic and organometallic compounds were made using $^1H$-$^1H$-gCOSY, $^1H$-$^{13}C$-HMBC, and $^1H$-$^{13}C$-HSQC NMR experiments. Abbreviations for NMR spectra are s (singlet), d (doublet), t (triplet), q (quartet), quint (quintet), sep (septet), dd (doublet of doublet), td (doublet of triplet), dq (doublet of quartet), m (multiplet), and br (broad). $^1H$ NMR signals of paramagnetic complexes are reported with line width at half-height ($\Delta v^{1/2}$) and chemical shift (δ), as the coupling constants are not measurable for these complexes. Air-sensitive NMR samples were prepared in a nitrogen glovebox using a standard NMR tube sealed with a septum and parafilm or a J. Young NMR tube.

**NMR measurement of paramagnetic complexes.** $^1H$ NMR spectra of paramagnetic complexes (typically 20 mM solution) were recorded with time domain data points (td) of 128 k, delay time (d1) of 0.1 s, scan number (ns) of 1–128, and spectral width (sw) of 200–600 ppm. Proton-coupled $^{13}C$ NMR spectra were recorded using an Avance NEO 500 spectrometer equipped with a cryoprobe. For proton-coupled $^{13}C$ NMR experiments, a pulse program of zg was used with time domain data points (td) of 64 k, delay time (d1) of 0.1 s, scan number (ns) of 64 k, and spectral width (sw) of 400–1100 ppm.

**Measurement of the effective magnetic moment by Evans' method.** Evans' method was conducted using glass capillaries containing deuterated solvent used to dissolve the NMR sample as external standards. The capillaries were inserted into the NMR tubes containing sample solutions, and $^1H$ NMR was measured before and after the insertion of the external standards to assign the signals from the standards.

**VSM measurement.** VSM measurements were conducted using the Quantum Design PPMS DynaCool VSM module. VSM powder sample holders (part #: 4096-388) were weighed outside the glovebox utilizing a microbalance and introduced into the glovebox. In a nitrogen glovebox, powdered samples were packed and sealed in the sample holders. The height of the packed samples was 1–2 mm. The sealed samples were taken out from the glovebox and weighed using a microbalance to calculate the weight of the samples.

**FTIR spectroscopy.** IR spectra were measured using a Nicolet iS5 FT-IR spectrophotometer outside the glovebox and are reported in frequency of absorption ($cm^{-1}$). Abbreviations for FT-IR spectra are s (strong), m (medium), and w (weak).

**UV-Vis spectroscopy.** UV-Vis spectra were measured using a Cary 60 UV-Vis spectrophotometer. The sample solution was added to a quartz cuvette (pathlength: 10 mm × 10 mm) in the glovebox, closed with a screw cap, and sealed with a parafilm, and the spectra were measured quickly outside the glovebox to avoid air oxidation of the samples.

**High-resolution mass spectrometry (HRMS).** HRMS were measured using a Bruker timsTOF or Thermo Scientific LTQ-Orbitrap mass spectrometers, using Electro Spray Ionization (ESI) mode.

**Elemental analyses.** Elemental analyses were carried out using an Exeter Analytical CE-440 Elemental Analyzer. Empty tin cups (EAI, Catalog #: 6703-0418) were weighed outside the glovebox utilizing a microbalance and introduced into the glovebox. In a nitrogen glovebox, 1–2 mg of samples were weighed and sealed. $N_2$ gas in the tin cups was replaced by argon by three vacuum argon refill cycles. The sealed samples were quickly weighed outside the glovebox using a microbalance to calculate the accurate weight of the samples and promptly placed in an autosampler to minimize the effect of air oxidation. All the samples were combusted and analyzed using the autosampler, which kept the samples in a He atmosphere.

## Data availability

All data are available in the main text or the supplementary materials. All data are available upon request from the corresponding author. Source Data are provided with this manuscript. Crystallographic data for the structures reported in this article and its supplementary information were deposited at the Cambridge Crystallographic Data Centre, under deposition numbers CCDC 2380049 (**1-H**), 2380050 (**1-Cl**), 2380051 (cocrystal of **1-OMe** and **2-OMe**), 2380052 (**2-OMe**), 2380053 (**2-NMe₂**), 2380054 (**3-H**), 2380055 (**3-Cl**), 2380056 (**3-OMe**), 2380057 (**3-NMe₂**), 2380058 (**4-H**), 2380059 (**4-NMe₂**), 2406042 (**4-NMe₂-PF₆**), 2406042 (**4-NMe₂-PF₆**), 2446847 (**Co-NMe₂**), 2446848 (**[Co-NMe₂]BF₄**), and 2446849 (**Ru-NMe₂**). Copies of the data can be obtained free of charge via https://www.ccdc.cam.ac.uk/structures/. Source data are provided with this paper.

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

## Acknowledgements

This work was supported by Japan Society for the Promotion of Science (JSPS), grant number 22K05134 (S.T.), JSPS Program for Forming Japan's Peak Research Universities, Okinawa Institute of Science and Technology Graduate University (OIST) instrumental analysis and engineering sections, and OIST Buribushi fellowship (S.T.). S.V.K. and R.R.F. conducted crystal structure determination and quantum topological analysis as part of the assignment to the FRC Kazan Scientific Center of RAS. The Mössbauer spectroscopic study was supported by Advanced Research Infrastructure for Materials and Nanotechnology (ARIM) of MEXT, Japan (no. JPMXP1223NI0406). We thank Prof. Urs Gellrich (University of Hohenheim) for his support with the DFT calculations.

## Author contributions

S.T. conceptualized the study, synthesized and characterized complexes, conducted SC-XRD measurements, organized collaborations, and prepared samples for Mössbauer spectroscopy, XPS, and VSM. J.A. conducted DFT calculations. R.R.F. and S.V.K. analyzed SC-XRD data and studied the quantum chemical topology. K.M. and T.O. conducted a Mössbauer spectroscopic study. H.-B.K. conducted VSM measurements. N.I. conducted XPS measurements. R.R.F. wrote the part describing the quantum topological study, and S.T. combined this part and wrote the original manuscript. S.T., J.A., S.V.K., R.R.F., and K.M. reviewed and edited the original manuscript.

## Competing interests

The authors declare no competing interests.
