## [Transparent Peer Review file · Nature Communications]

From 18- to 20-electron ferrocene derivatives via a ligand coordination

Corresponding Author: Dr Satoshi Takebayashi

Version 0:

Reviewer comments:

Reviewer #1

(Remarks to the Author)

This manuscript reports the synthesis of formal 20-electron ferrocene derivatives through reversible nitrogen coordination. While the experimental characterization is meticulous and technically sound, the novelty of the work is severely compromised by its heavy reliance on the authors' prior strategy for Co/Mn analog (Nat. Commun. 2023, 14, 4979). The authors' key innovation—introducing electron-donating groups (e.g., -NMe₂) to enhance pyridine coordination—is a logical extension of their prior work on Co/Mn, where the same ligand framework (CpNCp) was used to stabilize a 21-electron complex. This approach does not represent a paradigm shift but rather a predictable modification within an almost well-established design. In addition, while the data presentation and analyses are methodologically sound, the conclusions lack sufficient conceptual depth or broader implications. Therefore, I don't think this work in its current form is innovative enough to be published in a journal of Nat. Commun. as the previous paper, unless augmented by substantive advancements.

Here are some developments that transcend incremental progress and are likely to arouse broader interest:

1. Derivative Ligand Strategy with Conceptual Expansion. If the ligand design is generalizable, why not demonstrate its applicability to other metals (e.g., Ru, Mn⁺, oxidized Ni)? or provide unique insights on a direct comparison of Fe–N vs. Co–N bond orders, spin densities, or redox potentials
2. Exploration of "Tunability" in depth. Quantitative Structure-Activity Relationship (QSAR), such as how do Hammett parameters or DFT-calculated Lewis basicity correlate with equilibrium constants. or extreme examples: stronger electron donors, more polar solvents or even melts.
3. An enlightening theoretical framework. A comprehensive comparative DFT study is needed to highlight the uniqueness of Fe by comparing differences among various metals, and to more clearly elucidate how the environment and ligands can influence N-coordination.
4. Metal-Specific Applications: Demonstrate Fe's unique feature in catalysis (e.g., aerobic oxidation, C–H activation) using the 20-electron compound or its oxidized species.
5. Dynamic Coordination Chemistry: based on kinetics of N-coordination, proposing design rules for "switchable" electron counts (e.g., photo-/electro-triggered ligand binding).

Reviewer #2

(Remarks to the Author)

This work by Takebayashi and co-workers builds upon their fascinating work with "21 electron" cobaltocenes and the authors have expanded their efforts to thoroughly characterize some "20 electron" ferrocene derivatives. The authors use a large arsenal of spectroscopic, structural, and computational methods to strongly support their findings, which push the envelope of our understanding in the design of "hypervalent" organometallic complexes. Provided that some minor revisions are made, I support the acceptance of this manuscript into Nature Communications.

The most noteworthy features of this manuscript include the unusual S=2 electronic configuration adopted by inclusion of the amine donor. Moreover, the clear electronic dependence of the para substituent on its donor strength to iron is a nice touch to the overall narrative. Most complexes are impressively characterized using an effective blend of experimental and theoretical methods.

This work is similar to the author's previous work in the same journal on cobaltocene and manganese sandwich complexes (<https://www.nature.com/articles/s41467-023-40557-7>), however, this work is unique because it focuses exclusively on iron.

It also expands the field by a systematic modification of the ligand system to interrogate what differing the substituents para- to the donor nitrogen does to the complexation chemistry as well as the stability of the ligand.

I am overall quite pleased with the manuscript and its conclusions and claims, but there is some conjecture in the abstract and introduction that needs revision:

1. In the abstract, the authors say that their findings suggest “applications to unconventional catalytic reactions and organometallic materials”, however these claims are unsubstantiated and speculative when compared to the work presented in the manuscript. I recommend removing this, as it subtracts from the already interesting results in the main text.

2. In the introduction, the authors claim that “the stability of diamagnetic 18-electron complexes is a widely accepted fact.” This statement is overly broad, as there are probably thousands of 16-electron, square planar d8 systems that undergo ligand substitution reactions via kinetically and thermodynamically unstable 18-electron intermediates. The authors should reword this sentence, maybe clarifying that it is the stability of diamagnetic 18-electron complexes with d6 electron counts that are widely accepted as being kinetically and thermodynamically stable.

For 2-NMe₂ to 3-NMe₂, the Fe-N bond length from the Fe(II) to the Fe(III) increases by about 0.2 Angstroms and then contracts again by 0.3 Angstroms on oxidation to Fe(IV) (4-NMe₂). The contraction on going from Fe(III) to Fe(IV) is clear, but the initial extension of the Fe-N bond length upon oxidation is unexpected. This is pointed out in the text, however no clear explanation is given for this counter-intuitive result. Is this partly because of the shortened Fe—Cp distances that push away the pyridine ligand upon oxidation, or some other more subtle electronic reason? There is clearly a dramatic change in spin state from FeII to FeIII (S=2 to S = 1/2), so I would strongly recommend that the authors include spin density computations to probe the location(s) of the unpaired spin density, as the significant degree of Fe-N coupling in EPR spectra might be due to substantial ligand-based radical character.

A citation should be given for the Mossbauer data of ferrocene, as the authors clearly state that their complexes are “similar to ferrocene”. They may also want to consider adding this data to figure 3c for the sake of comparison.

On a separate note, there are a substantial number of references in the supporting material, and key references for the computational work are uncited in the main text. Some of these should be included in the main article (level of theory, quantum chemical topology, etc).

I believe that the methodology is sound and the authors exceed the expected standards, and there is a wealth of information in the SI on the preparation and characterization of these new compounds.

Reviewer #3

(Remarks to the Author)

Reviewer #4

(Remarks to the Author)

It is an excellent piece of work in which authors very convincingly proved that 2-NMe₂ is an 18-electron complex. The computational part is solid and I do not have any critical comment about this. I have only one suggestion: If the authors can also perform EDA-NOCV taking Fe as one fragment and the rest as another, and from the NOCV part they can understand the pair-wise orbital interaction energies for such 10 pair of electrons. This calculation is quite complicated but it could add really valuable information about the interaction energies in this intriguing complex.

Version 1:

Reviewer comments:

Reviewer #1

(Remarks to the Author)

The authors have addressed all my concerns, the paper is now acceptable.

Reviewer #2

(Remarks to the Author)

I now find this edited version acceptable for publication. Thank you for the detailed responses to all comments, congratulations on an excellent piece of work!

Note: I co-reviewed this manuscript with an Early Career Researcher. This is part of the Nature Communications initiative to

facilitate training in peer review and to provide appropriate recognition for Early Career Researchers who co-review manuscripts.

Reviewer #3

(Remarks to the Author)

Reviewer #4

(Remarks to the Author)

I am happy with the authors responses, and therefore, I recommend its publication in the current form.

Point-by-point response to the reviewers' comments

Reviewer #1 (Remarks to the Author):

This manuscript reports the synthesis of formal 20-electron ferrocene derivatives through reversible nitrogen coordination. While the experimental characterization is meticulous and technically sound, the novelty of the work is severely compromised by its heavy reliance on the authors' prior strategy for Co/Mn analog (Nat. Commun. 2023, 14, 4979). The authors' key innovation—introducing electron-donating groups (e.g., -NMe₂) to enhance pyridine coordination—is a logical extension of their prior work on Co/Mn, where the same ligand framework (CpNCp) was used to stabilize a 21-electron complex. This approach does not represent a paradigm shift but rather a predictable modification within an almost well-established design.

Response: Thank you very much for reviewing our manuscript. I agree that the key innovation in this paper is not the design of the ligands. The key innovation here is that 18-electron ferrocene derivatives can coordinate one more 2-electron donor and form formal 20-electron species. This coordination chemistry of ferrocene derivatives is entirely unexpected according to previous reports on ferrocene derivatives. More generally, according to the 18-electron rule, this type of coordination chemistry is not expected to happen with diamagnetic 18-electron complexes. We explained this point in the introduction section of the manuscript as "the coordination of a ligand to a diamagnetic 18-electron complex that results in a formal 20-electron complex is generally considered improbable even as a reaction intermediate⁸ and, to our knowledge, is unknown".

We previously reported the coordination of a 2-electron donor to 19-electron cobaltocene derivatives. This chemistry was also unexpected. However, the paramagnetic, 19-electron cobaltocene is one of the typical exceptions of the 18-electron rule. Thus, it is not significantly related to the 18-electron rule and its applications, such as catalysis. We mentioned this point in the manuscript as, "A previous study showed that the H-CpNCp ligand promotes nitrogen coordination to a derivative of paramagnetic 19-electron cobaltocene, a typical exception of the 18-electron rule". As such, even though we use the same ligand scaffold, the result presented in this work is not a logical extension of the previous work. Indeed, in the previous work, we showed that the 18-electron Co(3+) cobaltocenium complex did not form 20-electron species with the H-CpNCp ligand.

Apart from this point, the redox property of the Fe(CpNCp) complexes is a notable unique feature that no other ferrocene derivatives have and are not expected based on our previous work. In the previous work, we showed that our Co(CpNCp) cobaltocene derivative is less reducing than cobaltocene. In this study, we showed the Fe(CpNCp) derivatives are

significantly more reducing. It is so reducing that it can be oxidized twice with ferrocenium (FeCp_2^+). It was a completely unexpected result for us and will be so for most of the readers.

In addition, while the data presentation and analyses are methodologically sound, the conclusions lack sufficient conceptual depth or broader implications.

Response: Thank you for sharing your opinion. We included broader implications in the Electrochemical study section and the newly created section (Coordination chemistry of other 18-electron analogs).

Therefore, I don't think this work in its current form is innovative enough to be published in a journal of Nat. Commun. as the previous paper, unless augmented by substantive advancements.

Here are some developments that transcend incremental progress and are likely to arouse broader interest:

Response: Thank you for your suggestion. We included some of the suggested developments to increase broader interest.

1. Derivative Ligand Strategy with Conceptual Expansion. If the ligand design is generalizable, why not demonstrate its applicability to other metals (e.g., Ru, Mn+, oxidized Ni)?

Response: We synthesized neutral Ru(2+) and cationic Co(3+) complexes using $\text{NMe}_2\text{-CpNCp}$ ligand and made a new section to describe these results and interpretation of them. In both cases, the formation of the metal(M)-N bond was not observed in the solid state or in the solution. We propose that this is due to stronger, more stable M-Cp bonds between 2nd-row transition metal (Ru) or cationic (Co) complexes. The lower energy level of M-Cp bonding orbitals will increase the energy level of M-Cp antibonding orbitals. This will result in full (two electrons) occupation of the M-N antibonding orbital if the M-N bonding orbital is occupied. Therefore, the reported coordination chemistry of 18-electron complexes depends on the strength of the existing M-ligand bonds. If the existing M-ligand bonds are relatively weak, there may be an energetic advantage to form formal 20-electron complexes. Thus, neutral or anionic 1st-row transition metal complexes will have the highest chance to do this coordination chemistry, as metal-ligand bonds in these complexes tend to be weaker than those in 2nd and 3rd-row transition metals or cationic complexes.

or provide unique insights on a direct comparison of Fe-N vs. Co-N bond orders, spin densities, or redox potentials

Response: We computationally compared bond orders (Wiberg bond indices) of formal 19-electron Mn(2+), 20-electron Fe(2+), and 21-electron Co(2+) complexes using $\text{NMe}_2\text{-}$

CpNCp. We found that the Wiberg bond indices (Supplementary Table 8) of these complexes are virtually identical, as all of them have half-filled M-N antibonding orbitals. Consistent with this observation, Co(II) and Fe(II) complexes of NMe₂-CpNCp ligand have similar metal-N bond lengths.

2. Exploration of "Tunability" in depth. Quantitative Structure-Activity Relationship (QSAR), such as how do Hammett parameters or DFT-calculated Lewis basicity correlate with equilibrium constants. or extreme examples: stronger electron donors, more polar solvents or even melts.

Response: The DFT-calculated Gibbs free energy differences (in toluene at 298 K) of **1-X** and **2-X** and effect of solvent on this ΔG are already available in the reviewed SI, Supplementary Table 4 and Supplementary Fig. 32. We added Supplementary Fig. 58 to show correlation of Hammett *para*-substitution constants and the DFT-calculated enthalpy of Fe-N bond formation, and Fig. S59 to show correlation between the DFT-calculated Gibbs free energy differences of **1-NMe₂** and **2-NMe₂** versus dipole moments of solvent molecules. We also tried to determine the ratio of **1-NMe₂** and **2-NMe₂** in DMSO-d₆ using ¹H NMR. The faint pink color of the DMSO-d₆ solution indicates the predominant presence of **2-NMe₂** (**2-NMe₂** is an off-white solid). However, we could not determine the ratio of **1-NMe₂** and **2-NMe₂** as ¹H NMR signals from **1-NMe₂** and **2-NMe₂** could not be resolved at above 20 °C (melting point of DMSO-d₆). These experimental and theoretical studies support that more electron-donating substituents and more polar solvents favor the formation of Fe-N bonds.

3. An enlightening theoretical framework. A comprehensive comparative DFT study is needed to highlight the uniqueness of Fe by comparing differences among various metals, and to more clearly elucidate how the environment and ligands can influence N-coordination.

Response: These points (effect of metal, ligand substituents, and solvent) are clarified in the responses to the suggestion 1 and 2.

4. Metal-Specific Applications: Demonstrate Fe's unique feature in catalysis (e.g., aerobic oxidation, C-H activation) using the 20-electron compound or its oxidized species.

Response: We understand that catalytic application increases broader interest. However, this study is designed to reveal and understand the basic coordination chemistry of 18-electron ferrocene derivatives. This basic knowledge can be used to develop a new catalyst. In the newly created section (Coordination chemistry of other 18-electron analogs), we

included an interpretation of our result that may be usable for developing a catalytic reaction or for understanding existing catalytic reactions.

5. Dynamic Coordination Chemistry: based on kinetics of N-coordination, proposing design rules for "switchable" electron counts (e.g., photo-/electro-triggered ligand binding).

Response: We included possible applications of this chemistry in the Electrochemical study section and the newly created section (Coordination chemistry of other 18-electron analogs).

Reviewer #2 (Remarks to the Author):

This work by Takebayashi and co-workers builds upon their fascinating work with "21 electron" cobaltocenes and the authors have expanded their efforts to thoroughly characterize some "20 electron" ferrocene derivatives. The authors use a large arsenal of spectroscopic, structural, and computational methods to strongly support their findings, which push the envelope of our understanding in the design of "hypervalent" organometallic complexes. Provided that some minor revisions are made, I support the acceptance of this manuscript into Nature Communications.

The most noteworthy features of this manuscript include the unusual $S=2$ electronic configuration adopted by inclusion of the amine donor. Moreover, the clear electronic dependence of the para substituent on its donor strength to iron is a nice touch to the overall narrative. Most complexes are impressively characterized using an effective blend of experimental and theoretical methods.

This work is similar to the author's previous work in the same journal on cobaltocene and manganese sandwich complexes (<https://www.nature.com/articles/s41467-023-40557-7>), however, this work is unique because it focuses exclusively on iron. It also expands the field by a systematic modification of the ligand system to interrogate what differing the substituents para- to the donor nitrogen does to the complexation chemistry as well as the stability of the ligand.

I am overall quite pleased with the manuscript and its conclusions and claims, but there is some conjecture in the abstract and introduction that needs revision:

Response: Thank you very much for reviewing our manuscript.

1. In the abstract, the authors say that their findings suggest "applications to unconventional catalytic reactions and organometallic materials", however these claims are unsubstantiated

and speculative when compared to the work presented in the manuscript. I recommend removing this, as it subtracts from the already interesting results in the main text.

Response: Thank you for the suggestion. We agree that the statement is speculative compared to the presented work. We removed this statement.

2. In the introduction, the authors claim that "the stability of diamagnetic 18-electron complexes is a widely accepted fact." This statement is overly broad, as there are probably thousands of 16-electron, square planar d8 systems that undergo ligand substitution reactions via kinetically and thermodynamically unstable 18-electron intermediates. The authors should reword this sentence, maybe clarifying that it is the stability of diamagnetic 18-electron complexes with d6 electron counts that are widely accepted as being kinetically and thermodynamically stable.

Response: Thank you for pointing out this. We agree that there are unstable 18-electron complexes/intermediates. We changed the statement to:

"However, the stability of diamagnetic 18-electron complexes, especially complexes with d⁶ electron configuration, is the commonly accepted experimental foundation of this rule."

For 2-NMe₂ to 3-NMe₂, the Fe-N bond length from the Fe(II) to the Fe(III) increases by about 0.2 Angstroms and then contracts again by 0.3 Angstroms on oxidation to Fe(IV) (4-NMe₂). The contraction on going from Fe(III) to Fe(IV) is clear, but the initial extension of the Fe-N bond length upon oxidation is unexpected. This is pointed out in the text, however no clear explanation is given for this counter-intuitive result. Is this partly because of the shortened Fe—Cp distances that push away the pyridine ligand upon oxidation, or some other more subtle electronic reason? There is clearly a dramatic change in spin state from FeII to FeIII (S=2 to S = 1/2), so I would strongly recommend that the authors include spin density computations to probe the location(s) of the unpaired spin density, as the significant degree of Fe-N coupling in EPR spectra might be due to substantial ligand-based radical character.

Response: Thank you for pointing out this. In **4-H** and **4-NMe₂**, both Fe-Cp and Fe-N distances are shorter than those of **3-H** and **3-NMe₂**. Thus, "shortened Fe—Cp distances that push away the pyridine ligand upon oxidation" appear to not explain this observation. We compared the spin density of **2-NMe₂** and **3-NMe₂** (Supplementary Fig. 62). **2-NMe₂** showed the presence of significant spin density on Cp carbons (0.50 in total), and on pyridyl nitrogen atoms (0.04). In contrast, **3-NMe₂** showed the absence of spin density on Cp carbons and the presence of larger spin density on the pyridyl nitrogen (0.08). Consistent with these

observations, MO analysis of **3-NMe₂** (Supplementary Fig. 63) showed the unpaired electron occupies an orbital with Fe-N antibonding character. Therefore, the elongation of the Fe-N bond in **3-NMe₂** can be explained by the presence of a larger extent of unpaired spin density in **the** Fe-N antibonding orbital.

A citation should be given for the Mossbauer data of ferrocene, as the authors clearly state that their complexes are "similar to ferrocene". They may also want to consider adding this data to figure 3c for the sake of comparison.

Response: Thank you very much for pointing out this. We added a citation for the Mossbauer data of ferrocene in the text and in the caption of Fig. 3c. We chose relatively recent data measured at 90K (reported in 1999) where isomer shift is reported relative to that of metallic iron to compare with our data. The data is listed in Fig. 3c. The isomer shift and quadrupole splitting of ferrocene are temperature-dependent. However, the difference at 77 K and 90 K is expected to be small enough ($\sim 0.005 \text{ mm s}^{-1}$ according to the cited literature) for the accurate comparison of the data.

On a separate note, there are a substantial number of references in the supporting material, and key references for the computational work are uncited in the main text. Some of these should be included in the main article (level of theory, quantum chemical topology, etc).

Response: Thank you very much for pointing out this. We moved some of the citations in supporting material to the main text.

I believe that the methodology is sound and the authors exceed the expected standards, and there is a wealth of information in the SI on the preparation and characterization of these new compounds.

Reviewer #3 (Remarks to the Author):

Response: Thank you very much for reviewing our manuscript.

Reviewer #4 (Remarks to the Author):

It is an excellent piece of work in which authors very convincingly proved that 2-NMe₂ is an 18-electron complex. The computational part is solid and I do not have any critical comment about this. I have only one suggestion: If the authors can also perform EDA-NOCV taking Fe as one fragment and the rest as another, and from the NOCV part they can understand the pair-wise orbital interaction energies for such 10 pair of electrons. This calculation is quite complicated but it could add really valuable information about the interaction energies in this intriguing complex.

Response: Thank you very much for reviewing our manuscript. Unfortunately, the software we work with for the QM computations does not support EDA-NOCV.